

# The influence of pet ownership on self-compassion among nurses: a cross-sectional study

Hu Jiang[1,2], Yongxia Mei[1], Xiaoxuan Wang[1], Wenna Wang[1], Beilei Lin[1], Zhixin Zhao[1] and Zhenxiang Zhang[1]

[1] School of Nursing and Health, Zhengzhou University, Zhengzhou, Henan, China
[2] Nursing Department, The Third Affiliated Hospital of Zunyi Medical University (The First People's Hospital of Zunyi), Zunyi, Guizhou, China

## ABSTRACT

**Background.** The modern lifestyle trend of pet ownership is undoubtedly beneficial for both physical and mental health. Research has shown a connection between pet ownership and staff self-compassion. However, there has not been any evidence linking pet ownership to self-compassion in the nurse population.

**Aims.** To investigate the current status of pet ownership among nurses and explore the influence of pet ownership on self-compassion among nurses.

**Methods.** An online survey was conducted in July 2022 with 1,308 nurses in China. Data were collected using a general information questionnaire and a self-compassion scale. To compare categorical variables, the independent $t$ test, one-way ANOVA, and multiple linear regression analysis were utilized. SPSS software was used for the statistical analysis.

**Results.** We found that 16.9% of nurses owned at least one pet, and dogs and cats were the primary pets. The $t$ test for independent samples showed that pet owners and non-pet owners scored differently on self-compassion ($t = 3.286$, $p = 0.001$), self-kindness ($t = 3.378$, $p = 0.001$), common humanity ($t = 2.419$, $p = 0.016$), and mindfulness ($t = 2.246$, $p = 0.025$). One-way ANOVA revealed that the highest degree was an influencing factor of self-compassion ($\chi^2 = 1.386$, $p = 0.019$). Multiple linear regression showed that average monthly income, pet ownership, and highest degree were the factors that influenced self-compassion most significantly ($F = 8.335$, $p < 0.001$).

**Conclusion.** The results revealed that nurses actually own pets as part of their modern lifestyle, which provides them with social support and potentially enhances their self-compassion. More efforts should be focused on the impact of pet ownership on nurses' physical and mental health, and pet-based interventions should also be developed.

# INTRODUCTION

Nursing is a highly stressful career, and nurses have a high risk of burnout from a global perspective. Nurses are the main health force in the fight against COVID-19, but they face relatively high levels of stress (*Galanis et al., 2021*). A cross-sectional survey showed that 68.3% of nurses in China had high levels of occupational stress (*Gu, Tan & Zhao,*

Corresponding authors
Yongxia Mei, myx@zzu.edu.cn
Zhenxiang Zhang,
zhangzx6666@126.com

*2019*). Higher stress among nurses leads to anxiety, depression, burnout, and compassion fatigue and results in low job performance, reduced compassionate practice, poor patient outcomes, and even increased turnover to the detriment of organizational development (*Bakhamis et al., 2019*; *Dall'Ora et al., 2020*; *Garcia et al., 2019*; *Hoff, Carabetta & Collinson, 2019*; *Jun et al., 2021*).

To date, a wide variety of interventions have been used to manage nurses' stress. A literature review (*Chesak et al., 2019*) revealed that various stress management interventions for nurses have been investigated, with most focusing on treating an individual rather than their environment. Psychological skills training and mindfulness-based and meditation training were the most commonly used interventions. In addition to the altered environment or care model, massage, energy therapy, and multimodal interventions, there are also other types of interventions. Currently, evidence shows that most treatments are directed at treating the individual. However, there is still a gap between environmental intervention knowledge and efficacy.

Self-care and social support are significant strategies for nurses to deal with stress. To cope with this issue, the vast majority of nurses use self-care behaviors. Self-compassion is a form of self-care and is a skill that can be learned (*Mills, Wand & Fraser, 2018*). In recent years, self-compassion in particular has received attention from scholars. Self-compassion is a protective factor against stress. Individuals also require compassion for themselves, particularly during difficult times or when they are suffering (*Neff, 2003a*; *Neff, 2003b*).

*Neff (2003a)* and *Neff (2003b)* defined self-compassion as a regulation strategy in which feelings of worry or stress are not avoided in favor of being open and sensitive to one's own suffering, experiencing feelings of care and kindness to oneself, taking an attitude of understanding and not judging one's own inadequacies and failures, and acknowledging that one's own experience is part of the shared human experience. In 2016, *Neff (2016)* redefined self-compassion as a dynamic system that represents a synergistic state of interaction among the key elements of self-kindness, self-judgment, common humanity, isolation, mindfulness, and overidentification. A scoping review showed that self-compassion is helpful for reducing work-based stressors such as anxiety and compassion fatigue among nurses (*Steen, Javanmard & Vernon, 2021*). Despite the fact that nurses are charged with improving health and well-being, many do not practice compassionate self-care because of their busy schedules, reluctance, or limited competency (*Andrews, Tierney & Seers, 2020*; *Egan et al., 2019*).

Although self-compassion is a relatively new field, it has received increasing attention among nurses. Self-compassion can shield nurses from stress and burnout for their own health and possibly for the health of their patients. It was found that self-compassion can improve the job satisfaction and performance of nurses (*Chu, 2017*). Nurses often experience emotional exhaustion and are prone to job burnout and even compassion fatigue due to prolonged exposure to pain, but those with higher levels of self-compassion can generate compassion satisfaction, thereby reducing compassion fatigue (*Mooney et al., 2017*). To enhance nurses' self-compassion, interventions such as compassion-focused therapy (*Corrigan et al., 2022*) and mindful self-compassion (*Bodini et al., 2022*) have

been used. Recent research suggests that pet ownership may be a new strategy to foster self-compassion.

People are better able to deal with stress when they have pets as social support (*Barker et al., 2020*; *Krause-Parello, 2012*). A recent survey indicated that pet attachment could alleviate job insecurity and stress (*Wan et al., 2022*). According to *Clark, Smidt & Bauer (2018)*, weekly visits from a therapy dog may help reduce stress and feelings of being exhausted, frustrated, and overwhelmed that are related to working in the nursing industry. Research on nurses related to pet ownership or pet-assisted therapy is still relatively limited.

To date, pet ownership is the most common form of human-animal interaction (*Scoresby et al., 2021*). A survey conducted in the United States estimated that approximately 67% of households have at least one pet, with cats and dogs being the most popular choices (*Applebaum, Peek & Zsembik, 2023*). Chinese research showed that the number of pet-owning households reached 99.78 million in 2018, and the proportion of pet-owning households is approximately 17% (*Chen, 2019*), which is much lower than in other countries. Using attachment theory, experts suggest that people and animals form attachments to one another (*Westgarth et al., 2010a*). Moreover, pets are commonly recognized as family members and friends by most people (*Westgarth et al., 2010a*).

The connection between pet ownership and health has been researched more thoroughly in recent years. Because pet ownership provides social support, it has good physical and mental health-promoting effects. Pets can connect with nature, play a role in recreational and work activities, and provide companionship in the home. Pet-owning populations are more affectionate toward their neighbors, exercise more, and sleep better (*Mein & Grant, 2018*). Among elderly people, pet ownership contributes to successful aging, improves cognitive function, and enhances the quality of life among chronic patients (*Friedmann et al., 2020*; *Krittanawong et al., 2020*). Some studies have shown that pet companionship is a nonpharmacological approach to treating neurological disorders (*Boldig & Butala, 2021*). Pet ownership reduces the occurrence of psychological problems such as anxiety and depression, and pets may promote mental health recovery by creating a sense of identity (*Friesinger, Birkeland & Thorod, 2021*). A number of studies have indicated that pet ownership continues to have a positive psychological impact during the COVID-19 pandemic. *Oliva & Johnston (2021)* proved that pet ownership reduced stress and decreased the incidence of autism. *Tan et al. (2021)* suggested that pet ownership is beneficial for physical and mental well-being during periods of social isolation amidst a global pandemic.

Previous studies show that self-compassion and pet ownership are positively correlated. Companion animals help increase owners' self-compassion (*Kogan et al., 2021*). Research has demonstrated that veterans' self-compassion significantly improves with dog ownership in combination with a structured dog training program (*Bergen-Cico et al., 2018*). However, it is unclear how self-compassion and pet ownership relate to one another in the nursing population.

Despite promising initial findings on pet ownership related to self-compassion, to our knowledge, there are no published empirical studies on the status of pet ownership among Chinese nurses. Therefore, the present study aimed to investigate the current status of

pet ownership among nurses and to explore the relationship between pet ownership and self-compassion.

## MATERIALS AND METHODS

### Study design

A cross-sectional online survey of mainland Chinese nurses was conducted using convenience sampling. The present study adhered to the Strengthening the Reporting of Observational Studies in Epidemiology (STROBE) statement and checklist.

### Ethical considerations

This study was approved by the Ethics Committee of the First People's Hospital of Zunyi (2022-016), and permission for data collection was obtained from the participants. Prior to completing the survey, the participants were informed of the main content and the purpose of the study, and the participants could choose to give informed consent before participating in the survey. The participants were also informed that the results of the study would be presented in an aggregate form and that there was no way to identify the individuals who completed the survey. If a participant declined to participate, the questionnaire collection was automatically closed. Data and written informed consent were both recorded online.

### Sample size estimation

The sample size was calculated using PASS (Power Analysis and Sample Size) software 15.0. According to a previous study, approximately 17% of Chinese people owned pets (*Chen, 2019*), and when a two-sided $\alpha = 0.05$ with confidence level as 0.95 was set, a sample size of 906 was calculated. Considering a 10% questionnaire error rate, at least 997 nurses would need to be surveyed. In the end, 1,409 nurses responded to the questionnaire.

### Participants

All registered nurses who met the inclusion criteria were eligible to participate in this study. The participants had to meet the following inclusion criteria: (1) having worked as a nurse in a medical institution for more than one year and (2) being willing to participate in this study. The exclusion criteria were as follows: (1) stopped working as a nurse more than six months before the survey and (2) diagnosed with a severe psychological or physical disorder.

### Data collection

Data collection was carried out between 6 and 27 July 2022. Wenjuanxing (http://www.wjx.cn), a widely accepted online questionnaire survey platform in China for data collection, was used to construct a web questionnaire. Using a poster, the link and QR code of the web questionnaire were distributed *via* WeChat, a popular social media platform among users, and it was clearly indicated on the poster who would be included and excluded. Nurses could fill out the questionnaire by clicking on the link or scanning the QR code.

## Measurements
### The general information questionnaire

To collect general information, the authors designed a questionnaire based on the literature and the purpose of the study. This questionnaire mainly included demographic and sociological information such as gender, age, education, years of work, marital status, and ownership of pets and their types.

### Self-compassion scale

The Self-compassion Scale (SCS) was developed by *Neff (2003b)*. The Chinese version of the SCS that was used in this study was translated and revised by *Chen, Yan & Zhou (2011)*). The scale has 26 items in six dimensions: self-kindness, common humanity, mindfulness, self-judgment, isolation, and overidentification. The items included in self-judgment, isolation, and overindulgence should be scored in reverse. The Cronbach's $\alpha$ coefficient of the total scale was 0.80, and the Cronbach's $\alpha$ coefficients of each subscale were 0.73, 0.70, 0.77, 0.77, 0.77, and 0.70, respectively. The score of each dimension is the average score of all items in the dimension, and the total score of self-compassion is the sum of the average scores of each dimension. It is scored using the Likert 5-point system. The authors have permission to use this instrument from the copyright holders (*Chen, Yan & Zhou, 2011*).

## Statistical analyses

IBM SPSS 26.0 software was used for data analysis. Continuous variables were reported as the mean ($\pm$ standard deviation [SD]), and the count data were described as frequency and percentage. The Kolmogorov–Smirnov and Shapiro Wilk tests were used to check the normality of the data. Independent $t$ tests, one-way ANOVA and multiple linear regression analysis were used to analyze the influencing factors of self-compassion. A difference's magnitude was measured by the effect size. We used Cohen's d value to determine the effect size in the t test. We considered $d = 0.20$ small, $d = 0.50$ medium, and $d = 0.80$ large. Cohen's f value was used for the analysis of variance, and we considered small, medium, and large effect sizes of $f = 0.10$, $f = 0.25$, and $f = 0.40$, respectively. Data analysis was performed using the least significant difference (LSD) post hoc method to compare the groups. $p < 0.05$ was considered indicative of statistical significance.

## RESULTS
### Descriptive statistics

A total of 1,409 nurses responded to the questionnaire, of which 58 declined to participate in this survey, resulting in 1,351 questionnaires being received. Furthermore, 42 questionnaires were judged to be invalid due to missing or incorrect information. The final 1,309 questionnaires were included in the analysis, with a valid return rate of 92.9%. The participants' ages ranged from 18 to 59 years old, and the average age was $32.25 \pm 6.18$. A total of 1267 (86.7%) of the participants were female, and 42 (3.2%) were male. Some of the nurses owned pets, and 1088 nurses did not own pets. Dogs and cats were the most common sorts of pets kept (55.7% and 52.5%, respectively), followed by other animals (15.4%), with some people having both kinds of pets, and most pet owners (76.0%) having had their animals for longer than a year. The results are shown in Table 1.

**Table 1 Demographic characteristics of participants (N = 1,309).**

| Demographics | n (%) |
|---|---|
| Age (years) | |
|     <30 | 444 (33.9) |
|     30~ | 724 (55.3) |
|     40~ | 112 (8.6) |
|     50~ | 29 (2.2) |
| Gender | |
|     Male | 42 (3.2) |
|     Female | 1267 (96.8) |
| Personality Type | |
|     Extrovert | 577 (44.1) |
|     Introvert | 732 (55.9) |
| Highest Degree | |
|     Junior college and below | 200 (15.3) |
|     Bachelor's | 1075 (82.1) |
|     Master's and above | 34 (2.6) |
| Sibling | |
|     Yes | 159 (12.1) |
|     No | 1150 (87.9) |
| Marital status | |
|     Unmarried | 337 (25.7) |
|     Married | 949 (72.5) |
|     Others | 23 (1.8) |
| Children | |
|     Yes | 430 (32.8) |
|     No | 879 (67.2) |
| Job title | |
|     None | 85 (6.5) |
|     Junior Level | 672 (51.3) |
|     Intermediate level | 502 (38.3) |
|     Senior level | 50 (3.8) |
| Working years (years) | |
|     ≤5 | 348 (26.6) |
|     6–15 | 770 (58.8) |
|     ≥16 | 191 (14.6) |
| Average monthly income (yuan) | |
|     ≤3000 | 141 (10.8) |
|     3001~6000 | 725 (55.4) |
|     6001~9000 | 311 (23.8) |
|     ≥9001 | 132 (10.1) |

**Table 1** (*continued*)

| Demographics | *n* (%) |
|---|---|
| Number of night shifts per month | |
| 0 | 439 (33.5) |
| 1–4 | 238 (18.2) |
| ≥5 | 632 (48.3) |
| Pets | |
| Dog | 123 (55.7) |
| Cat | 116 (52.5) |
| Others | 34 (15.4) |
| Length of pet ownership | |
| ≤ 6 months | 31 (14.0) |
| 6~12 months | 22 (10.0) |
| >12 months | 168 (76.0) |

**Table 2**  Scale scores between pet-owners and non-pet owners ($N = 1,309$).

| Scores | Pet-owners ($n = 221$) | Non-pet owners ($n = 1088$) | *t* | *p* | Effect size |
|---|---|---|---|---|---|
| Self-compassion | 81.89 ± 13.20 | 78.92 ± 12.05 | 3.286 | 0.001 | 0.24 |
| Self-judgment | 13.40 ± 4.03 | 13.16 ± 3.65 | 0.832 | 0.406 | 0.06 |
| Isolation | 11.07 ± 3.91 | 10.65 ± 3.17 | 1.479 | 0.140 | 0.13 |
| Over-identification | 11.74 ± 3.64 | 11.30 ± 3.11 | 1.687 | 0.093 | 0.14 |
| Self-kindness | 17.89 ± 3.57 | 17.01 ± 3.51 | 3.378 | 0.001 | 0.25 |
| Common humanity | 13.46 ± 2.92 | 14.34 ± 3.20 | 2.419 | 0.016 | −0.28 |
| Mindfulness | 14.34 ± 3.20 | 13.85 ± 2.91 | 2.246 | 0.025 | 0.17 |

### Scale scores between pet owners and non-pet owners

Using independent samples, t tests showed that pet owners scored differently than non-pet owners on self-compassion ($t = 3.286$, $p = 0.001$), self-kindness ($t = 3.378$, $p = 0.001$), common humanity ($t = 2.419$, $p = 0.016$), and mindfulness ($t = 2.246$, $p = 0.025$). However, there were no differences between the two groups for self-judgment ($t = 0.832$, $p = 0.406$), isolation ($t = 1.479$, $p = 0.140$), and overidentification ($t = 1.687$, $p = 0.093$). The results are shown in Table 2.

### Univariate analysis and multiple linear regression of self-compassion

One-way ANOVA revealed that the highest degree was an influencing factor of self-compassion ($\chi^2 = 1.386$, $p = 0.019$). The LSD post hoc analysis showed that nurses with junior college and below degree had a lower level of self-compassion than those with a bachelor's degree ($p = 0.010$) and a master's and above degree ($p = 0.022$) had, but there was no significant difference between those with a bachelor's degree and a master's and above degree ($p = 0.194$). Furthermore, there were no statistically significant differences among the other variables ($p > 0.05$). The results are presented in Table 3. In a multiple linear regression using the forward method, self-compassion was used as the dependent variable, and demographic characteristics were used as the independent variable. The

findings showed that average monthly income, pet ownership, and highest degree were the factors that influenced self-compassion most significantly ($F = 8.335$, $p < 0.001$). The results are shown in Table 4.

## DISCUSSION

In this study, we found that 16.9% (221) of the nurses owned at least one pet. We also discovered that pet owners had higher self-compassion than non-pet owners. In addition, average monthly income, pet ownership, and highest degree were influencing factors of self-compassion. The results revealed that pet ownership may promote nurses' self-compassion.

To the best of our knowledge, pet ownership among nurses in mainland China has not been reported by other studies. This cross-sectional study showed that 16.9% of the nurses owned at least one pet, and this rate was similar to the data reported in a previous study (*Chen, 2019*). Currently, little research has been conducted on the status of individual group pet ownership. The literature on nurse pet ownership on a national and international scale is lacking, and this study fills this gap. Our research shows that pet ownership among Chinese nurses is on par with that of the general population. The rate of pet ownership in China is much lower than that abroad. Consistent with previous studies, dogs and cats were the primary pets (*Applebaum, Peek & Zsembik, 2023*). Nevertheless, there are large differences between cultures and types of household pets, with dogs dominating in some countries and cats in others (*Fraser et al., 2020*; *Kim & Chun, 2021*), and there was a small difference between cats and dogs in our current study. Additionally, many people owned at least one type of pet, and most nurses owned pets for at least a year. Although studies have shown that pet ownership is a social act for individuals and is influenced by subjective and objective factors (*Purewal et al., 2019*; *Westgarth et al., 2010b*), there is a lack of research regarding the predictors of pet ownership among nurses. Thus, more studies are needed to examine these predictors.

Another finding of our study was that pet ownership is also associated with self-compassion. In this study, the results showed that pet owners had a higher level of self-compassion than non-pet owners, which is consistent with previous findings (*Bergen-Cico et al., 2018*; *Kogan et al., 2021*). Pet owners had a higher level of self-kindness, common humanity, and mindfulness than non-pet owners, which indicated a positive effect of pet ownership on self-compassion. High levels of self-compassion can promote self-care behaviors, which further promote individuals' physical and mental health. However, the relationship among pet ownership, self-compassion, and physical and mental health was not examined in depth in this study and needs to be further explored in future studies.

Moreover, we found that pet ownership was also a predictor of self-compassion, and nurses who keep pets had a higher degree of self-compassion in the multiple linear regression. This could be because pet ownership can reduce individuals' anxiety and stress (*Bolstad et al., 2021*; *Brooks et al., 2018*) and increase positive feelings of happiness and security (*Brooks et al., 2018*), which may be a strategy for coping with stress. Pet ownership also provides social support for owners (*Antonacopoulos & Pychyl, 2010*; *Applebaum et*

**Table 3 Univariate analysis of Self-compassion ($N = 1{,}309$).**

| Variables | Self-compassion | $\chi^2/t$ | $p$ | Effect size |
|---|---|---|---|---|
| Age | | 0.798 | 0.892 | 0.19 |
| <30 | 79.48 ± 13.03 | | | |
| 30~ | 79.21 ± 11.96 | | | |
| 40~ | 79.49 ± 12.35 | | | |
| 50~ | 83.62 ± 7.96 | | | |
| Gender | | −0.915 | 0.360 | −0.14 |
| Male | 77.71 ± 11.83 | | | |
| Female | 79.48 ± 12.31 | | | |
| Personality Type | | −0.728 | 0.467 | −0.04 |
| Extrovert | 79.14 ± 12.61 | | | |
| Introvert | 79.64 ± 12.05 | | | |
| Highest Degree | | 1.386 | 0.019 | 0.28 |
| Junior college and below | 77.29 ± 13.50 | | | |
| Bachelor's | 79.72 ± 12.01 | | | |
| Master's and above | 82.50 ± 12.53 | | | |
| Sibling | | 0.060 | 0.316 | 0.08 |
| Yes | 80.34 ± 11.08 | | | |
| No | 79.30 ± 12.46 | | | |
| Marital status | | 1.098 | 0.272 | 0.11 |
| Unmarried | 79.47 ± 12.59 | | | |
| Married | 79.33 ± 12.18 | | | |
| Others | 82.35 ± 12.99 | | | |
| Children | | −0.716 | 0.474 | 0.00 |
| Yes | 79.59 ± 13.02 | | | |
| No | 79.59 ± 11.93 | | | |
| Job title | | 1.092 | 0.281 | 0.24 |
| None | 77.92 ± 15.17 | | | |
| Junior Level | 79.57 ± 12.34 | | | |
| Intermediate level | 79.11 ± 11.87 | | | |
| Senior level | 83.16 ± 9.80 | | | |
| Working years (years) | | 0.787 | 0.906 | 0.13 |
| ≤5 | 79.74 ± 12.62 | | | |
| 6–15 | 79.07 ± 12.54 | | | |
| ≥16 | 80.26 ± 10.62 | | | |
| Average monthly income (yuan) | | 1.071 | 0.323 | 0.35 |
| ≤3000 | 78.12 ± 14.29 | | | |
| 3001 ~6000 | 78.64 ± 12.60 | | | |
| 6001 ~9000 | 81.08 ± 11.00 | | | |
| ≥9001 | 81.20 ± 10.62 | | | |
| Number of night shifts per month | | 0.707 | 0.971 | 0.04 |
| 0 | 79.36 ± 12.99 | | | |
| 1–4 | 79.71 ± 11.84 | | | |
| ≥5 | 79.36 ± 11.98 | | | |

**Table 4  Multiple linear regression of self-compassion.**

| Variables | B | SD | β | t | p |
|---|---|---|---|---|---|
| Constant | 76.764 | 2.938 | | 26.125 | < 0.001 |
| Average monthly income | 0.984 | 0.446 | 0.064 | 2.206 | 0.028 |
| Highest Degree | 1.928 | 0.881 | 0.063 | 2.189 | 0.029 |
| Pet ownership | −2.826 | 0.904 | −0.086 | −3.127 | 0.002 |

**Notes.**

$R^2 = 0.019$, adjusted $R^2 = 0.017$, $F = 8.335$, $p < 0.001$.

*al., 2020*). A longitudinal control study suggested that dog ownership in combination with a structured dog training program could increase self-compassion among veterans (*Bergen-Cico et al., 2018*).

According to attachment theory, pet attachment is a specific form of attachment that also occurs in adults (*Westgarth et al., 2010a*). Attachment behavior refers to a set of emotional and behavioral strategies that are connected to close relationships and interactions with others (*Antonucci, Akiyama & Takahashi, 2004*). In adulthood, attachment is classified according to two dimensions: attachment avoidance and attachment anxiety. According to *Ding & Xu (2021)*, self-compassion moderates the relationship between attachment anxiety and self-esteem and has a mediating effect on subjective well-being. The findings in this study support the idea that pet ownership is associated with self-compassion, but the mechanisms by which pet attachment and self-compassion interact need to be further investigated.

In addition, average monthly income and education were also associated with levels of self-compassion, and this finding is consistent with previous studies. A longitudinal follow-up study reported that higher income is associated with higher self-compassion levels (*Lee et al., 2021*). Another cross-sectional study showed that less-educated individuals had lower self-compassion (*López et al., 2018*). Although the association among these factors is quite complex, we hypothesize that people with higher incomes and education are more inclined to self-regulate to increase self-compassion, but more research is still needed to confirm this in the nurse population.

## Limitations

To our knowledge, this is the first study to examine the influence of pet ownership on nurses' self-compassion. However, this study has several limitations. Our survey was conducted online, and the participants might be more limited by our research methodology, which does not yet fully represent Chinese nurses' overall pet ownership. Additionally, there are more women than men in nursing due to the specificity of the profession. This means that the results cannot be generalized to other populations. Furthermore, as our findings were based only on a status quo survey, the results are limited by the variables studied, and more long-term and in-depth studies are needed. Additionally, the degree of explanation of our model is relatively low, and further validation is needed.

## CONCLUSION

As a result of this study, we found that 16.9% of nurses owned at least one pet, which revealed that nurses actually own pets as part of their modern lifestyle, providing them with social support and potentially also enhancing their self-compassion. In the future, more empirical studies are needed to confirm the correlation between pet ownership and nurses' physical and mental health, especially occupational health, psychological mechanisms, and short- and long-term effects. In addition, pet companionship can be explored as a new psychological intervention strategy.

### Funding

This study was funded by the National Natural Science Foundation of China (72174184), the Youth Project of National Natural Science Foundation of China (72104221) and the Science and Technology Fund Project of Guizhou Provincial Health and Health Commission (gzwjkj2019-1-030). The funders had no role in study design, data collection and analysis, decision to publish, or preparation of the manuscript.

### Grant Disclosures

The following grant information was disclosed by the authors:
The National Natural Science Foundation of China: 72174184.
Youth Project of National Natural Science Foundation of China: 72104221.
Science and Technology Fund Project of Guizhou Provincial Health and Health Commission: gzwjkj2019-1-030.

### Competing Interests

The authors declare there are no competing interests.

### Author Contributions

- Hu Jiang conceived and designed the experiments, performed the experiments, analyzed the data, prepared figures and/or tables, authored or reviewed drafts of the article, and approved the final draft.
- Yongxia Mei conceived and designed the experiments, performed the experiments, analyzed the data, prepared figures and/or tables, authored or reviewed drafts of the article, and approved the final draft.
- Xiaoxuan Wang conceived and designed the experiments, performed the experiments, analyzed the data, prepared figures and/or tables, authored or reviewed drafts of the article, and approved the final draft.
- Wenna Wang conceived and designed the experiments, performed the experiments, analyzed the data, prepared figures and/or tables, authored or reviewed drafts of the article, and approved the final draft.
- Beilei Lin conceived and designed the experiments, performed the experiments, analyzed the data, prepared figures and/or tables, authored or reviewed drafts of the article, and approved the final draft.

- Zhixin Zhao conceived and designed the experiments, performed the experiments, analyzed the data, prepared figures and/or tables, authored or reviewed drafts of the article, and approved the final draft.
- Zhenxiang Zhang conceived and designed the experiments, performed the experiments, analyzed the data, prepared figures and/or tables, authored or reviewed drafts of the article, and approved the final draft.

## Ethics

The following information was supplied relating to ethical approvals (i.e., approving body and any reference numbers):

This study was approved by the Ethics Committee of the First People's Hospital of Zunyi (2022-016).

## Data Availability

The raw measurements are available in the Supplemental File.

## Supplemental Information

Supplemental information for this article can be found online at http://dx.doi.org/10.7717/peerj.15288#supplemental-information.

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
