# Peer review of "The influence of pet ownership on self-compassion among nurses: a cross-sectional study"

_PeerJ, doi:10.7717/peerj.15288_

## Round 0.1 · original submission · Minor Revisions

Dear Author,

Please consider and address the comments from reviewers carefully.

Reviewer 1 ·

Basic reporting

overall, the article is very interesting and there is no research that discusses this. there are only a few notes. attached notes for author and editor.

Experimental design

the experimental design was clearly enough, please focus on ethical consideration. How did the researcher avoid bias in the selection of participants according to the inclusion and exclusion criteria?

Validity of the findings

Validity and reliability was done.

just complate the major finding on your research. there are only a few notes. attached notes for author and editor.

Annotated reviews are not available for download in order to protect the identity of reviewers who chose to remain anonymous.

Reviewer 2 ·

Basic reporting

no comment.

Experimental design

no comment.

Validity of the findings

no comment.

Additional comments

Introduction:
1. Please add the data of nurses experiencing stress in your research location.
2. Please add the efforts or intervention that have been made by the government, hospitals, nurse associations, or other research to reduce nurse stress levels and the results. So that pet ownership interventions can be unique and novelty in this study.
3. The author has explained the phenomenon well. Please include the theory or theoretical framework that underlies this research so that it becomes the basis for selecting variables and demographic characteristics of respondents.
Conclusion:
1. Please paraphrase the conclusion in abstract and in main text. The sentences in both sections are repetition.

Reviewer 3 ·

Basic reporting

1. The English language is straightforward, easy to understand.
2. The literature references supported the discussion points.
3. Professional article structure, figures, tables and raw data are shared.
4. Results are related to the hypothesis.

Experimental design

The methods are described for replication.

Validity of the findings

The benefit to the nurse population may be beneficial to China's context.
Conclusions are related to the findings, but may be general.

Additional comments

Referencing check, proofreading and copyediting are necessary.

---

## Round 0.2 · Minor Revisions

Dear author,
The manuscript is significantly improved. However, I have a concern with the typography such as capital in the first sentence, language and grammar, and the abbreviation (eg. PASS (Power Analysis and Sample Size) --> should be Power Analysis and Sample Size (PASS)). Please read the manuscript carefully to revise it and you can send the manuscript for professional English editing (can be proven by certificate).

---

## Round 0.3 · accepted · Accept

Thank you for the revised manuscript and English correction done with certification